# Direct observation of elemental fluctuation and oxygen octahedral distortion-dependent charge distribution in high entropy oxides

Lei Su[1,6], Huaixun Huyan[1,6], Abhishek Sarkar [2,3], Wenpei Gao [1], Xingxu Yan [1], Christopher Addiego [4], Robert Kruk[3], Horst Hahn [2,3] & Xiaoqing Pan [1,4,5]

The enhanced compositional flexibility to incorporate multiple-principal cations in high entropy oxides (HEOs) offers the opportunity to expand boundaries for accessible compositions and unconventional properties in oxides. Attractive functionalities have been reported in some bulk HEOs, which are attributed to the long-range compositional homogeneity, lattice distortion, and local chemical bonding characteristics in materials. However, the intricate details of local composition fluctuation, metal-oxygen bond distortion and covalency are difficult to visualize experimentally, especially on the atomic scale. Here, we study the atomic structure-chemical bonding-property correlations in a series of perovskite-HEOs utilizing the recently developed four-dimensional scanning transmission electron microscopy techniques which enables to determine the structure, chemical bonding, electric field, and charge density on the atomic scale. The existence of compositional fluctuations along with significant composition-dependent distortion of metal-oxygen bonds is observed. Consequently, distinct variations of metal-oxygen bonding covalency are shown by the real-space charge-density distribution maps with sub-ångström resolution. The observed atomic features not only provide a realistic picture of the local physico-chemistry of chemically complex HEOs but can also be directly correlated to their distinctive magneto-electronic properties.

[1] Department of Materials Science and Engineering, University of California, Irvine, CA 92697, USA. [2] KIT-TUD-Joint Research Laboratory Nanomaterials, Technical University Darmstadt, 64287 Darmstadt, Germany. [3] Institute of Nanotechnology, Karlsruhe Institute of Technology, 76344 Eggenstein-Leopoldshafen, Germany. [4] Department of Physics and Astronomy, University of California, Irvine, CA 92697, USA. [5] Irvine Materials Research Institute, University of California, Irvine, CA 92697, USA. [6]These authors contributed equally: Lei Su, Huaixun Huyan. ✉email: horst.hahn@kit.edu; xiaoqinp@uci.edu

igh-entropy oxides (HEOs) are a rapidly emerging group of chemically complex functional materials, which encompasses single-phase solid solutions containing five or more principal cations[1–5]. The key strength of HEOs is the superior compositional flexibility that allows the stabilization of numerous chemical compositions with varying crystallographic structures[6–14]. Consequently, diverse and appealing functional properties are exhibited by HEOs, such as high ionic conductivity[15], superior durability for the applications in energy storage, conversion and catalysis[16–21], sluggish thermal conductivity with improved modulus[22,23], and exotic magnetic phenomena[24]. In general, the elucidation of the functional properties in oxides requires information about the intricate detail of local structural and chemical bonding characteristics, such as short-range elemental distribution, the metal-oxygen bond angle variations, extent of metal-oxygen orbital overlap and in effect bond covalency. On the one hand, realistic experimental visualization of these local aspects is challenging and the task becomes more daunting in the case of HEOs with their chemical complexity. On the other hand, the fundamental understanding of local atomistic features is pivotal in any effort to optimize the properties of HEOs, thus, necessitating the application of innovative experimental techniques to get insight into the very local physico-chemistry.

The recently developed real-space charge-density mapping with sub-ångström resolution utilizing four-dimensional (4D) scanning electron diffraction performed in a state-of-the-art aberration-corrected scanning transmission electron microscope (AC-STEM) offers an experimental platform to reconstruct the local atomic characteristics in systems with the highest precision[25–28]. In contrast to atom probe tomography, which also provides superior atomic resolution, the charge-density mapping by STEM diffraction imaging offers the possibility to unveil the detail of chemical bonding that is crucial in ionic systems with widely varying degree of covalency. In addition, the AC-STEM also provides atomic-scale elemental distribution to reveal local chemical ordering, which is found to be of paramount importance for HEAs[25] but is rarely reported in HEOs.

In this work, the correlation between local elemental distribution, atomic structure, chemical bonding, and magnetic properties in a series of model rare-earth (RE)-transition-metal (TM)-based perovskite-HEOs (P-HEOs) is investigated utilizing AC-STEM. P-HEOs inherit the base magnetic–electronic features of the conventional RE-TM perovskites. However, these are accompanied by a unique set of properties that presumably stem from the short-range elemental order and varying nature of the TM–O–TM bond characteristics. Three P-HEOs with a disordered TM based B-site are explored, with one system also having a disordered A-site: $La(Cr_{0.2}Mn_{0.2}Fe_{0.2}Co_{0.2}Ni_{0.2})O_3$, $Gd(Cr_{0.2}Mn_{0.2}Fe_{0.2}Co_{0.2}Ni_{0.2})O_3$ and $(La_{0.2}Nd_{0.2}Sm_{0.2}Gd_{0.2}Y_{0.2})(Cr_{0.2}Mn_{0.2}Fe_{0.2}Co_{0.2}Ni_{0.2})O_3$. The experimental visualization of the local atomistic features provided here (as schematically illustrated in Fig. 1) indicate obvious compositional fluctuations and composition-dependent variation of oxygen octahedral distortion in P-HEOs. The real-space charge-density distributions between the metal and oxygen columns present a direct relation between the composition (Fig. 1), local oxygen octahedral distortion, and the extent of TM (3d)-O (2p) bond covalency. Consequently, a direct correlation between the experimentally evidenced local atomic bonding characteristics of HEOs and their magnetic properties, both generic, such as the dependency on magnetic transition temperatures, and unconventional ones, such as gradual magnetic phase transitions and magnetoelectronic phase separation, can be readily drawn.

## Results

**Preparation and crystal structure of the P-HEOs.** The P-HEOs and a representative RE-TM perovskite, $LaCoO_3$, for comparison, were prepared using the nebulized spray pyrolysis (NSP) method with a subsequent heat treatment process at 1200 °C in air[12]. The studied P-HEOs, $La(Cr_{0.2}Mn_{0.2}Fe_{0.2}Co_{0.2}Ni_{0.2})O_3$, $Gd(Cr_{0.2}Mn_{0.2}Fe_{0.2}Co_{0.2}Ni_{0.2})O_3$, and $(La_{0.2}Nd_{0.2}Sm_{0.2}Gd_{0.2}Y_{0.2})(Cr_{0.2}Mn_{0.2}Fe_{0.2}Co_{0.2}Ni_{0.2})O_3$, will henceforth be addressed as $La(5TM_{0.2})O_3$, $Gd(5TM_{0.2})O_3$, $(5RE_{0.2})(5TM_{0.2})O_3$, respectively. X-ray diffraction (XRD) patterns coupled with the Rietveld refinements are presented in Supplementary Fig. 1, which confirm the phase purity of these compounds. The $LaCoO_3$ sample exhibits a rhombohedral ($R$-$3c$) crystal structure, while all three entropy-stabilized perovskites are orthorhombic ($Pbnm$) in lattice symmetry. To illustrate the structural characters of P-HEOs quantitatively, we employed Goldschmidt's tolerance factor concept. An average ionic radius[8] was used to calculate the tolerance factor (which is referenced as mean tolerance factor, MTF) of the P-HEOs based on Eq. (1)[29]:

$$t = \frac{r_A + r_B}{\sqrt{2}(r_B + r_O)} \tag{1}$$

where $t$ is the mean tolerance factor, $r_A$, $r_B$, and $r_O$ are the ionic radii of A-site element, B-site element, and oxygen, respectively. Ideal perovskite materials have a cubic closed packed structure with a tolerance factor of 1. When the value of the tolerance factor deviates from 1, an internal strain and crystal distortion arises. The tolerance factor of $LaCoO_3$ is 0.98, and the MTF of $La(5TM_{0.2})O_3$, $(5RE_{0.2})(5TM_{0.2})O_3$, and $Gd(5TM_{0.2})O_3$ are 0.96, 0.92, and 0.90, respectively.

With the decrease of MTF, a significant decrease in the $a$ and $c$ lattice parameters in P-HEOs is observed (Supplementary Fig. 1). Likewise, with the decrease of the MTF, the superstructure reflections get stronger, suggesting an increasing degree of orthorhombic distortion from $La(5TM_{0.2})O_3$ to $(5RE_{0.2})(5TM_{0.2})O_3$ to $Gd(5TM_{0.2})O_3$. Supplementary Fig. 2 shows the structure models of the three P-HEOs obtained from the Rietveld refinement of the XRD pattern, from which the differences between the degree of the orthorhombic nature in the P-HEOs can be observed. In STEM imaging and EDS, [110] direction of the perovskite crystal is selected to easily differentiate oxygen atoms from metal atoms directly. From the annular bright-field (ABF) and high-angle annular dark-field (HAADF) images in Figs. 2a, b, 3a, b and Supplementary Fig. 3a, b, it is seen that all three P-HEOs show sharp lattice fringes, verifying their perfect crystallinity. In $LaCoO_3$, La and Co are uniformly distributed in both A and B-site of the $ABO_3$ structure, respectively, evident from the contrast difference between the large heavy atoms and the small light atoms in the ABF image in Supplementary Fig. 4a.

**Elemental fluctuation.** Figure 2a, d displays the HAADF image and EDS map of $La(5TM_{0.2})O_3$, respectively, confirming its chemical compositions. EDS maps show that La is uniformly distributed on the A-site, and the five transition-metal (TM) cations, namely Fe, Mn, Cr, Ni, and Co, are in the B-site. Oxygen is observed in the ABF image (Fig. 2b), distributed between two neighboring B-site atoms. The brightness of an individual spot represents the abundance of a specific element along the [110] zone axis. From the EDS maps in Fig. 2d, the intensity variations of the five B-site TM cations can be clearly observed. To visualize this composition variation, line profiles of the B-site elements in a (001) plane projected along [110] zone axis are presented. Figure 2e–h shows the intensity variation of the five transition metals along each line as shown in Fig. 1d, which indicates that

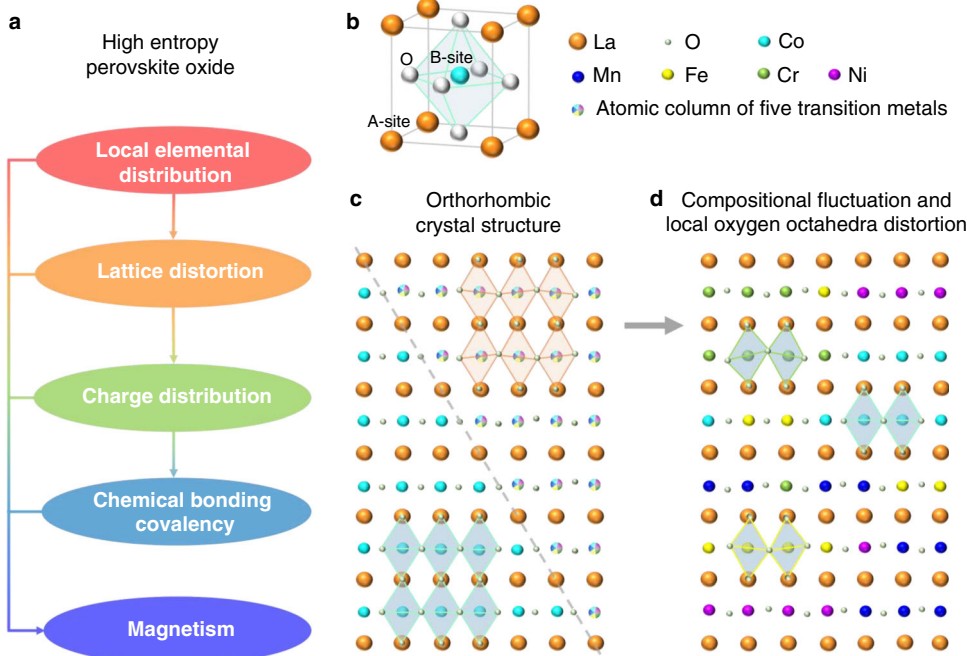

**Fig. 1 Illustration of the concept of the present study. a** The correlation between local elemental distribution, atomic structure, chemical bonding, and magnetic properties. **b** Schematic crystallographic structure of $LaCoO_3$. **c** Rhombohedral $LaCoO_3$ parent structure (bottom left) with negligible oxygen octahedral distortion and orthorhombic model of high-entropy oxide (top right) with uniformly distributed elements and oxygen octahedral distortion (as that derived from XRD pattern). **d** Compositional fluctuation with nanometer-scale ordering and composition-dependent local oxygen octahedral distortion of the high-entropy perovskite oxide that we observed in the present study. In (**c**) and (**d**), most of the oxygen atoms overlapped with A-site La atoms are omitted for visual clarity, and only those oxygen atoms belonging to the marked oxygen octahedra are present. Local chemical composition fluctuation of transition metal at each B-site atomic column gives rise to the different degree of local oxygen octahedral distortion, as indicated by the varied TM–O–TM bonding angle marked between the B-site atomic columns and the oxygen ones.

the atomic fraction of Co, Cr, Ni, Fe, or Mn in each projected atomic column randomly fluctuates with obvious variation and that elements can concentrate locally in nanometer regions. The degree of fluctuation in atomic occupancy among all atomic columns is different among these 5 TM elements. For example, the atomic fractions of Cr, Mn, Fe, Co, and Ni along the first column in Line 1 are 0.265, 0.205, 0.070, 0.318, and 0.142, respectively (Fig. 2e), while those along the first column in Line 2 are 0.269, 0.185, 0.263, 0.173, and 0.110, respectively (Fig. 2f). The atomic fraction of the TM cations between the neighboring columns changes irregularly (Fig. 2e–h), indicating no obvious preference for specific neighbors.

Similar to $La(5TM_{0.2})O_3$, the Gd occupies A-site uniformly, while fluctuations are seen in B-site among the 5 compound cations in the matrix of $Gd(5TM_{0.2})O_3$ (Supplementary Fig. 3c–g). In the case of $(5RE_{0.2})(5TM_{0.2})O_3$, the five RE metal cations, namely La, Nd, Y, Sm, and Gd, are located in the A-site and the above-mentioned five TM cations in B-site (Fig. 3a, c). All of these cation sites show a similar behavior with compositional fluctuation, evident from the brightness variation of each rare-earth elements and transition metals (Fig. 3c). The line profiles of the atomic intensity representing the distribution of individual A-site and B-site elements of $(5RE_{0.2})(5TM_{0.2})O_3$ in a (001) plane projected along the [110] zone axis are in Fig. 3d–f, g–i, respectively.

The appearance of these composition fluctuations is similar to the elemental distribution behavior in some high-entropy alloys[30,31]. It is worth noting that despite the above-mentioned strong composition fluctuations, local aggregated clusters with well-defined sizes and clear outlines cannot be readily identified. Therefore, the composition fluctuation observed here is more likely to be incipient concentration waves. This kind of local inhomogeneous elemental distribution thus can be an important

addition to the current common understanding that the constituent elements in non-metallic high-entropy materials, such as oxide[3,8], nitrides[18], carbides[32], and borides[33], exhibit long-range disorder distributions, which presents deep insights into the understanding of the atomic-scale elemental distribution in chemically complex systems.

**Tunable and broad distribution of oxygen octahedral distortion.** We then used ABF imaging, which is sensitive to oxygen atoms, to investigate the lattice distortion in the P-HEOs along [110] zone axis. As shown in Figs. 2b, 3b, 4a, and Supplementary Fig. 4, O atoms, which are located between the neighboring B-site elements, can be observed clearly in the ABF images of $LaCoO_3$ and the three entropy-stabilized perovskite oxides. We define the atom positions by finding mass center of each B-site and oxygen atom followed by Gaussian fitting, based on which, the TM-O-TM angles in the four samples were measured (Fig. 4b and Supplementary Table 1). In the case of $LaCoO_3$, almost all the O atoms are located at the normal position, presenting an average Co–O–Co angle of 177.9°, indicating that there is rarely lattice distortion. While for the three P-HEOs, obvious lattice distortion is observed (Fig. 4a). Compared to $La(5TM_{0.2})O_3$, $(5RE_{0.2})(5TM_{0.2})O_3$, and $Gd(5TM_{0.2})O_3$ show more pronounced oxygen octahedral tilting. As shown in Fig. 4b and Supplementary Table 1, the average TM-oxygen-TM (TM–O–TM) angles of $La(5TM_{0.2})O_3$, $(5RE_{0.2})(5TM_{0.2})O_3$, and $Gd(5TM_{0.2})O_3$ samples are measured as 175.6°, 153.4°, and 142.6°, respectively. Such distortion is related to the formation of perovskite structure and can be quantified by applying Goldschmidt's tolerance factor concept. We plot the MTF of the P-HEOs as a function of TM–O–TM angles in Fig. 4c. The angle decreases as the MTF decreases, in other words, the larger the MTF deviates from 1, the

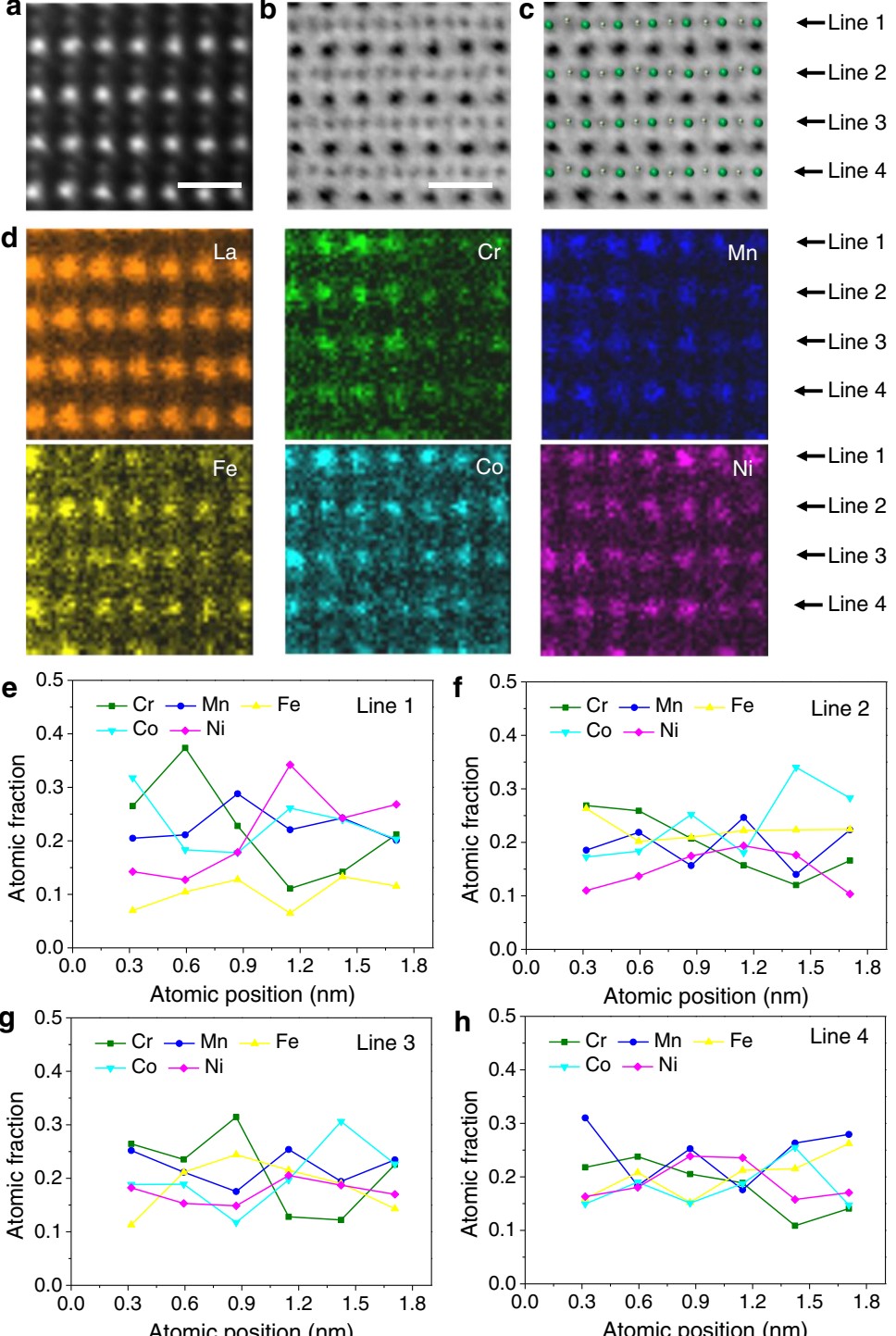

**Fig. 2 Atomic structure and compositional fluctuation of La(5TM0.2)O3. a** HAADF-STEM, **b** ABF, and (**c**) marked ABF images of La(5TM$_{0.2}$)O$_3$ taken from [110] zone axis. Scale bar, 0.5 nm. The green and silver dots marked in c denotes the B-site atoms and oxygen atom, respectively. **d** Atomic EDS mapping showing the disordered elemental distribution in the B-site of the ABO$_3$ structure. **e–h** Line profiles of the atomic fraction calculated from the relative intensity, representing the distribution of individual B-site elements in a (001) plane projected along the [110] zone axis.

more significant is the distortion of the oxygen octahedral. This trend in the variation of BO$_6$ distortion corresponds well to the crystal structure of P-HEOs derived from XRD patterns (Supplementary Fig. 2).

Although the structural features of the P-HEOs exhibit a proportion variation with respect to the tolerance factor, it is important to note that P-HEOs show a broad distribution of the

TM–O–TM bond angles (as shown in Fig. 4b) that is not seen in LaCoO$_3$. For example, in ABF image of La(5TM$_{0.2}$)O$_3$, the TM-oxygen octahedra (BO$_6$) is not uniformly distorted (Fig. 2b, c). Some of the unit cells have no oxygen tilt while in certain regions the oxygen octahedral distortion is observed, showing the distortion fluctuation in the P-HEOs and a broad distribution from about 173.2 to 178.0°. This can be considered as a direct

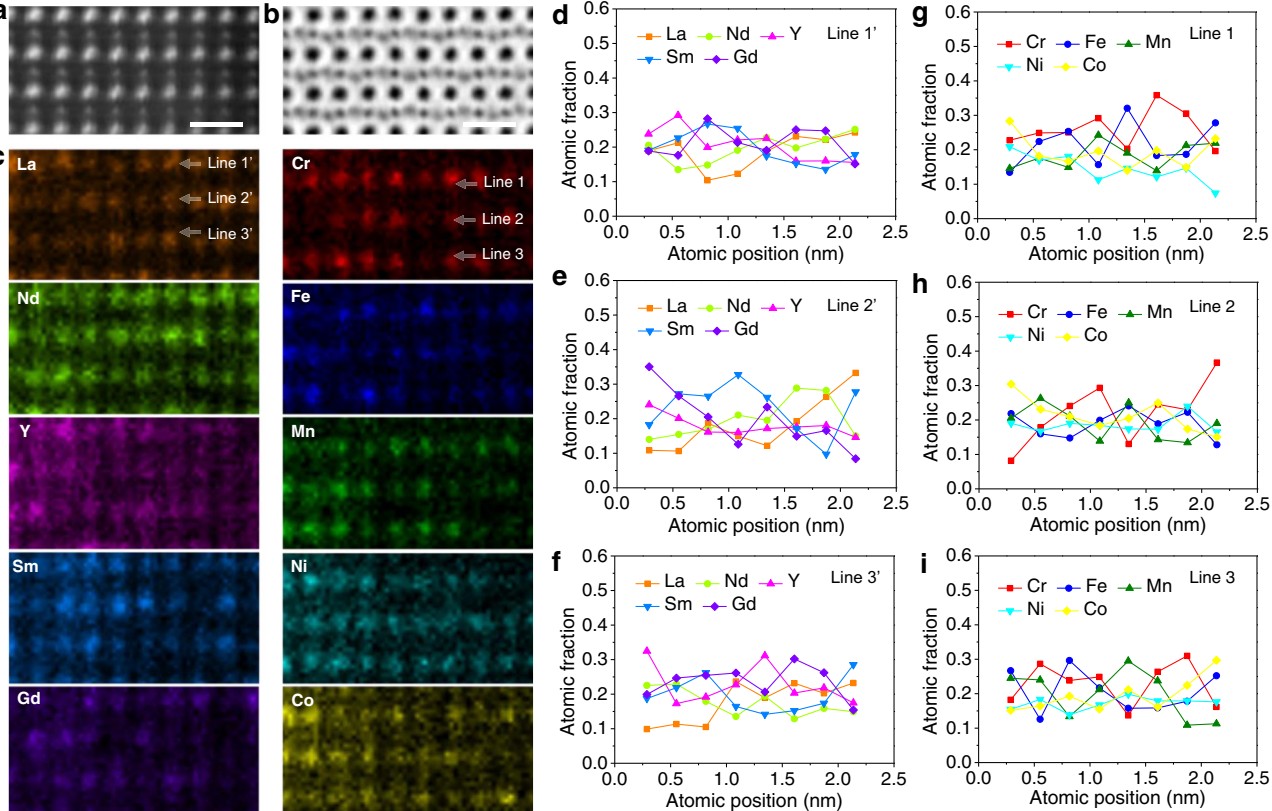

**Fig. 3 Atomic structure and compositional fluctuation of (5RE0.2)(5TM0.2)O3. a** HAADF-STEM and **b** ABF images of $(5RE_{0.2})(5TM_{0.2})O_3$ taken from [110] zone axis. Scale bar is 0.5 nm. **c** Atomic EDS mapping showing the disordered elemental distribution in the B-site of the $ABO_3$ structure. Line profiles of the atomic fraction calculated from the relative intensity fraction, representing the distribution of individual (**d**–**f**) A-site and (**g**–**i**) B-site elements in a (001) plane projected along the [110] zone axis.

consequence of the variety of TM cations present in P-HEOs with different ionic radii as well as the inhomogeneous distribution of internal stress in the matrix of P-HEOs. Compared with $La(5TM_{0.2})O_3$, in the case of $(5RE_{0.2})(5TM_{0.2})O_3$, after introducing more RE elements as A-site elements, a more complicated lattice mismatch and strain distribution formed. While in the case of $Gd(5TM_{0.2})O_3$, after replacing La with Gd in the 6-cations perovskite system, the MTF becomes smaller and the internal stress in the matrix becomes larger. Therefore, the distribution ranges of TM–O-TM angle in both the two cases (141.2–172.7° and 127.5–154.6° for $(5RE_{0.2})(5TM_{0.2})O_3$ and $Gd(5TM_{0.2})O_3$, respectively) are largely extended (Fig. 4b). Although the oxygen octahedral distortion-induced zig-zag bond stacking can be observed from the structure obtained from the Rietveld refinements of the XRD pattern (Supplementary Fig. 2), it should be noted that XRD provides only an average value of the TM–O–TM bond fluctuation. Hence, in the case of P-HEOs, containing several TM cations (with different ionic radii) resulting in different TM–O–TM bonds, the local picture of the bond stacking observed through ABF imaging becomes important.

Such local oxygen octahedral distortions have also been observed in simple perovskite oxides in previous studies through pair distribution function analysis of the neutron diffraction results[34–36]. These local octahedral distortions show a close association with the physical properties of perovskite oxides. For instance, the interaction between local distortions induces dipole moments that could form a spontaneous polarization, thus resulting in ferroelectricity in $BaTiO_3$[35]. Another example is $La_{0.60}Ca_{0.40}MnO_3$, where the local distorted oxygen octahedra results in the local anisotropic environment around $Mn^{4+}$, which

leads to a small degree of local ferroelectricity without any obvious macroscopic polarity[36]. Therefore, direct observation of local oxygen octahedral distortion and unraveling the relationship between chemical compositions and octahedral distortion in the P-HEOs would facilitate the designing and engineering of their local lattice structure and physical properties.

**Oxygen octahedral distortion-dependent charge distribution and bond covalency.** To provide insight into the relationships between the distortion and the electronic properties, real-space charge mapping of the four samples was then conducted by 4D STEM[25]. As illustrated in Supplementary Fig. 5 and Supplementary Methods, the thickness of the samples used for 4D-STEM analysis is below 6.6 nm, indicating that the samples are within the acceptable thickness range for charge mapping[25]. As is shown, the charge maps contain negative contributions (indicated as red in Fig. 5a) from both core and valence electrons and positive contributions (indicated in blue in Fig. 5a) from atomic nuclei. In the charge-density maps of $LaCoO_3$ and the three P-HEOs, the rare-earth metal column, transition-metal column, oxygen column as well as the severe oxygen octahedral distortion in the P-HEO samples are all clearly observed, which provides a fascinating perspective to analyze the relationship between chemical bonds and oxygen octahedral distortion in the P-HEOs. In the case of $LaCoO_3$, the charge-density map shows a nearly 180° TM–O–TM angle, in agreement with ABF measurements. While in the case of $La(5TM_{0.2})O_3$, due to the decrease of the tolerance factor, the O columns deviate from the standard position, resulting in the decrease of the TM–O–TM angle. Further

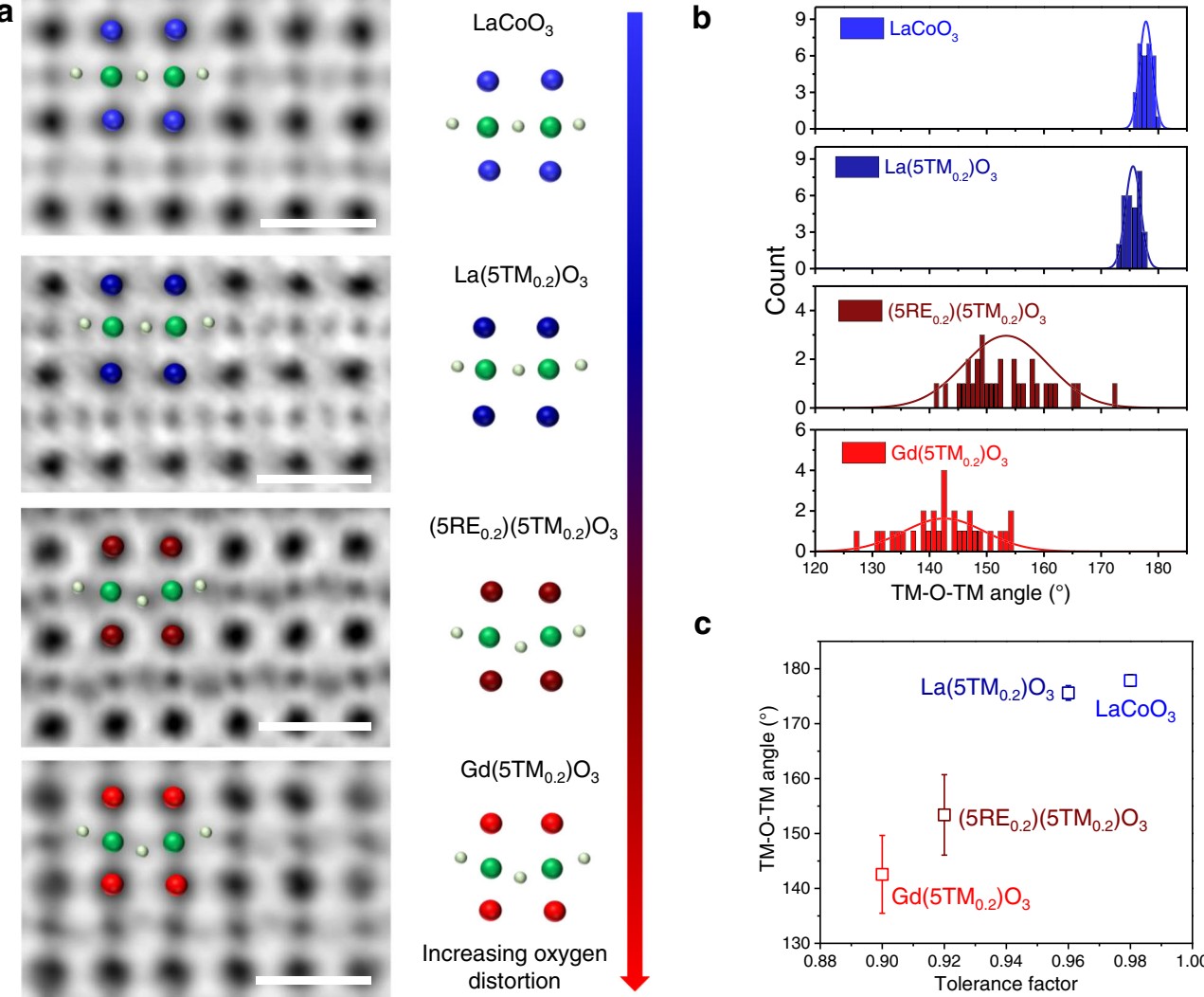

**Fig. 4 Oxygen tilt of the four samples. a** ABF images of $LaCoO_3$, $La(5TM_{0.2})O_3$, $(5RE_{0.2})(5TM_{0.2})O_3$, and $Gd(5TM_{0.2})O_3$, respectively, showing the TM–O–TM angle in their lattice structures. The green and silver spheres representing the B-site atoms and oxygen atoms, respectively. While the blue, dark blue, dark red and red spheres representing the A-site atoms in the four samples, respectively. Scale bar is 0.5 nm. **b** Distribution of the TM–O–TM angle in the four samples. **c** The relationship between TM–O–TM angle and tolerance factor. Data are presented as means of ±SD of the distribution of TM–O–TM angle.

decrease of the MTF results in the decrease of the TM–O–TM angle. These phenomena correspond well to the TM–O–TM angle as indicated in the ABF images in Fig. 4a.

It is worth noting that the charge-density distribution shows an interesting evolution with the decrease of the TM–O–TM angle. In $LaCoO_3$, each atom is clearly surrounded by regions of negative charge, meaning there are shared electrons in the interatomic regions (Fig. 5a); this implies stronger covalency in Co–O bonding. In comparison, with the increase of oxygen octahedral distortion in the P-HEO samples, the negative charge pockets between the O column and TM column become weaker, suggesting the decrease of electron charge density and thus a decrease of the covalency of the TM–O bond[25]. Considering covalent bonding in transition-metal perovskite oxides often takes place via hybridization between *3d* TM and *2p* O orbitals, this could suggest a lowering of the strength of the hybridization of the *3d* TM and *2p* O orbitals, which is of utmost importance governing several physical properties in oxide systems, like the optical, electronic and magnetic.

To provide further insight into the relationships between the chemical bond and lattice distortion, electron energy-loss

spectroscopy (EELS) analysis was then conducted. Figure 5b shows O *K*-edge and $L_{2,3}$ peaks of all the other elements except Y ($L_{2,3}$ edges of Y are over 2000 eV). In conventional RE-TM perovskites, the lowest-energy O *K*-edge feature typically corresponds to the *3d* TM and *2p* O orbital overlap[37–39]. Gaussian fitting is applied to find these peak positions, revealing that the significant difference between the 4 spectra is in a peak region from 528 to 534 eV. The lowest-energy O *K*-edge peaks of $LaCoO_3$, $La(5TM_{0.2})O_3$, $(5RE_{0.2})(5TM_{0.2})O_3$, and $Gd(5TM_{0.2})O_3$ are located at 531, 531.7, 531.9, and 533 eV, respectively, showing a right shift. The shifts may stem from the changing of TM–O-TM bonding geometry or transition-metal oxidation state, or both[37]. In previous work, researchers calculated the energy-loss near-edge structures of $LaFeO_3$ and $Bi_{0.9}La_{0.1}FeO_3$ by density functional theory[38]. Because the electronegativity of La (1.10) is only about half of that of Bi (2.02), less electrons are expected on the Fe and O sites in the case of $Bi_{0.9}La_{0.1}FeO_3$, resulting in the right shift of the lowest-energy O K-edge peak. In another case, for a series of Mn oxides, the O K-edge increases from 528.02 to 532.00 eV when the Mn ions state changes from +4 to +2[40]. These examples have shown that the onset position of O K-edge

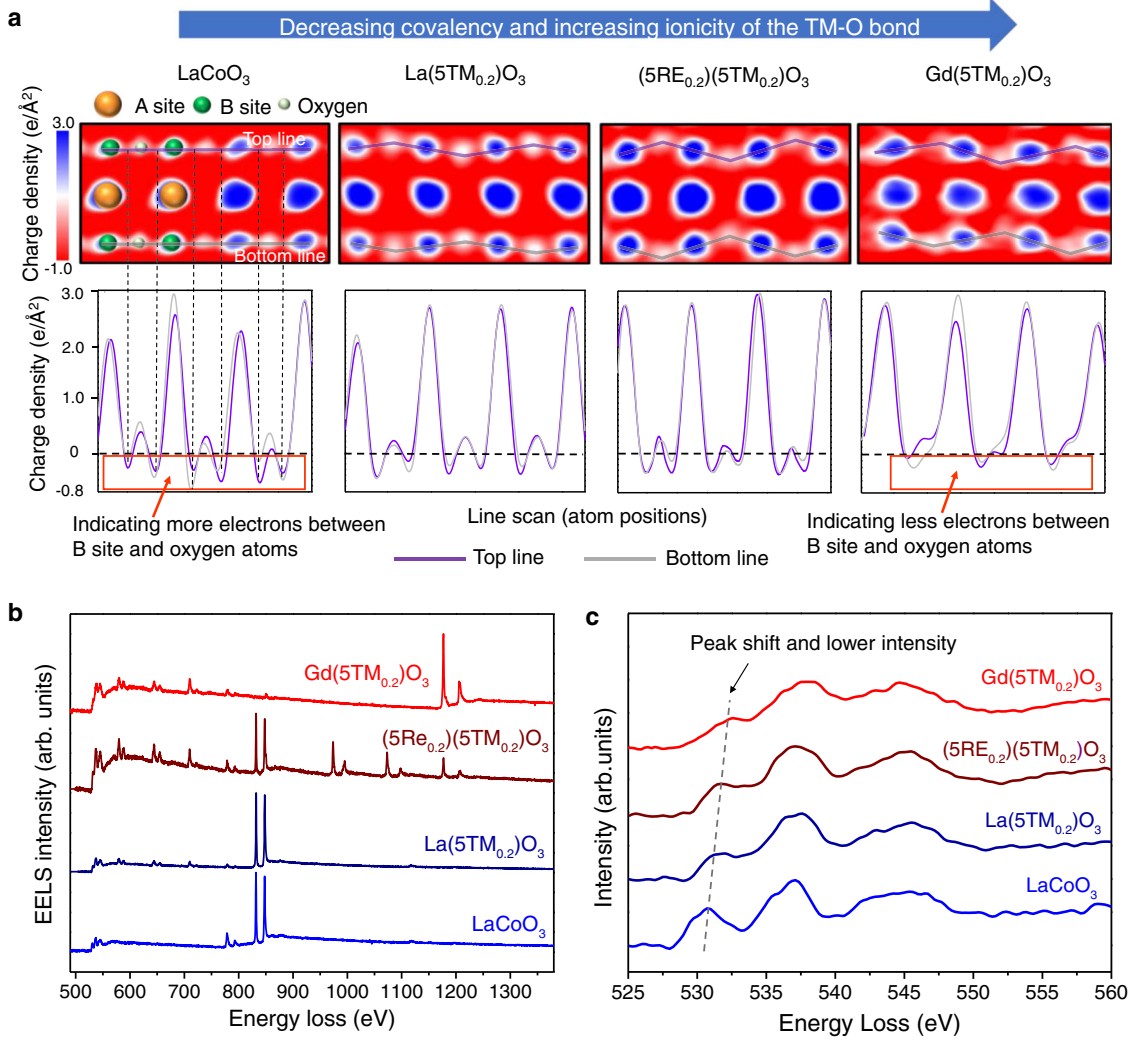

**Fig. 5 Real-space charge mapping and EELS analysis of LaCoO₃ and the three P-HEOs. a** Charge mapping of the three P-HEOs, indicating a decrease of the covalency and increase of the ionicity of TM–O bond. The colored lines in the line profiles of the charge distribution show the variation along the lines (Top line and Bottom line) as indicated in the charge maps. **b** EELS of LaCoO₃ and the three P-HEOs. **c** Detailed EELS of O *K*-edge electron energy-loss near-edges structures of the four samples. The dash line in (**c**) is a guide to eye showing the right shift of the lowest-energy O *K* peaks.

increases with the decrease of TM oxidation state. For the present samples, the states of the RE cations in the A-site and oxygen ions are +3 and −2, respectively, and the electronegativity of these RE elements (1.10, 1.14, 1.17, 1.20, and 1.22 for La, Nd, Sm, Gd, and Y, respectively) is very close from one to another. Consequently, the average oxidation state of the TM cations in the B-site does not change. Therefore, the shift of the O K-edge might only stem from the difference in the geometry of the TM–O–TM bond in the P-HEOs. Previous STEM, EELS, and first-principles band structure calculation study on the electronic structure of layered double perovskite La₂CuSnO₆ has shown that oxygen octahedral distortions in the material would lead to an expansion of the Cu–O–Cu bond, resulting in the shift of the O K-edge position at the Cu site to a higher energy[41]. Another research on the electronic structure of perovskites CaCu₃Ir₄O₁₂, CaCu₃Ru₄O₁₂, and CaCu₃Ti₄O₁₂ also shows that the different B-site elements in the three oxides result in the difference of the TM–O (Cu–O and Ca–O) bond lengths, thus influencing the interaction between TM and oxygen[42]. Therefore, the onset of O K-edge of CaCu₃Ti₄O₁₂ is located at the highest energy position among the three perovskite oxides. In the present case, with the increase of the TM–O–TM bond angles in the P-HEOs, the bond length of

the TM–O–TM expands[41]. Consequently, the O K-edge position shift to higher energy position. Besides the peak shift, the intensity of the lowest-energy O *K*-edge peaks of P-HEOs also shows significant decrease when compared with that of LaCoO₃, which indicates a decrease of the covalency of the TM–O bond[37,43–46], corresponding well to the findings as indicated in charge maps.

**Correlation between the crystallographic structure and magnetic properties**. The crystallographic structure and the related TM–O–TM bond angle, constituent TM elements, TM-oxygen octahedral distortion and the extent of the 3*d* TM and 2*p* O overlap are of great significance for determining the magnetic properties of perovskite-type oxides. In previous work, it was postulated that the magnetic ordering of La(5TM₀.₂)O₃, which sets in gradually below 185 K, originates from the competing interactions between the predominant antiferromagnetic (AFM) matrix and nanometer length-scale weak ferromagnetic (FM) domains[24]. The presence and coupling between the two magnetic phases, enveloped in a single crystallographic structure, also resulted in an observable vertical exchange bias (VEB)[24]. Speculations were made about the presence of short-range FM

correlations stemming from the preferential ferromagnetic exchange interactions between some of transition-metal cations.

Magnetic exchange interactions in ionic compounds, which in case of the studied P-HEOs is predominantly superexchange antiferromagnetic (AFM), depend on the overlap of exponentially decaying orbital wave functions of the nearest-neighbor cations, in $3d$-transition metals (TM) the $3d$ orbitals, that interact through the filled $2p$-orbitals of the neighboring oxygen. The exchange interaction between the participating ions in a simplified form can be described by the Heisenberg Hamiltonian, $H$ (as indicated in Eq. 2), where $J_{ab}$ is the exchange parameter coupling the two cations $a$ and $b$, whose spins are $S_a$ and $S_b$, respectively.

$$H = -2J_{ab}S_aS_b \qquad (2)$$

The summation of all such binary metal ion interactions via oxygen (for instance in the frame of a mean-field theory) determines the magnetic state of the material. The exchange parameter $J_{ab}$ directly depends on the degree of hybridization or, in other words, on the orbital overlap between the neighboring ions. In the class of materials considered here, the strength of hybridization is directly correlated with the bond angles. An increase in the degree of hybridization means an increase in the covalent nature of the bond, which results in a stronger superexchange interaction. Consequently, stronger superexchange interactions lead to higher transition temperatures (Néel-temperature: $T_N$). Extensive studies on conventional orthorhombic RE-TM perovskites indicate that the interaction parameter $J_{ab}$ is proportional to the cosine of the TM–O-TM bond angle ($\theta$). Initially, it was proposed that $J_{ab}$ is proportional to $\cos(\theta)$, while later studies reported that $\cos^2(\theta)$ directly provided a linear dependency of $T_N$ [47–49]. The most recent studies propose a modified correlation, i.e., $J_{ab}$ is proportional to $\cos^4(\omega/2)/d^7$ where $\omega = 180°-\theta$ and $d = $ TM–O bond length [47]. Given the value $d$ is not constant in P-HEOs, as in single B-site perovskites, we have plotted $T_N$ as a function of $\cos^2(\theta)$ in Supplementary Fig. 6. As illustrated by the charge-density maps and EELS analysis, the deviation of the TM–O–TM bond angle from 180° indicates lesser orbital overlap or decrease in the strength of hybridization. Hence, as displayed in Table 1, a decrease in the $T_N$ from La(5TM$_{0.2}$)O$_3$ to (5RE$_{0.2}$)(5TM$_{0.2}$)O$_3$ to Gd(5TM$_{0.2}$)O$_3$ (from 185 K to 135 K to 120 K, respectively) is in complete agreement with the changes in the average TM–O-TM bond angles (175.6°, 153.4°, and 142.6°, respectively), alterations indicated by the zig-zag lines in the charge-density maps and the redshift in the lowest-energy feature of the O $K$ ELNES feature (531.7, 531.9, and 533.0 eV, respectively) directly observed in this work.

Essentially, the magnetic characteristics of P-HEOs can be broadly classified into two categories. One is the generic feature, such as the onset temperature of magnetic phase transition, which is strongly dependent on the overall crystallographic structure, the average bond angle, the zig-zag alteration in the charge-

density features, and the resulting extent of the $3d$ TM and $2p$ O hybridization. On the other side, the distinctive magnetic features, such as the gradual magnetic phase transition distributed over a range of temperatures and magnetic phase separation resulting in VEB are largely dictated by the local structural features, for instance, the broad distribution of TM–O-TM bond angles and the elemental fluctuations in P-HEOs. The presence of the VEB necessarily indicates a magnetic phase separation in crystallographic single-phase P-HEOs. While the superexchange AFM interactions are dominant in P-HEOs, it is plausible that the presence of the FM phase/clusters is related to the preferential ferromagnetic exchange between some of the TM cations. Although a definitive assignment the FM interactions to certain local atomistic feature is not straightforward, the current experimental observation of elemental fluctuation on the TM B-sublattice in P-HEOs can be considered as one of the main factors (Figs. 2 and 3). Likewise, the gradual onset of the magnetic transition over a large temperature regime can also be related to the chemically complex structure of the P-HEOs. The strength of the exchange interactions, which decides the magnetic transition temperature, are locally different in P-HEOs depending upon the local chemical compositions (Figs. 2 and 3) and broad distribution in the bond angles (Fig. 4b, c). Hence, instead of a sharp transition, a gradual onset of magnetic ordering is observed in the P-HEOs. Altogether, by utilizing the strength of the detailed STEM atomic-scale characterization techniques, it can be shown that the unique magnetic properties of P-HEOs are a direct consequence of their, especially local, structure and electronic distinctiveness.

## Discussion

In conclusion, we experimentally show the relationship between distinct magnetic phenomena observed in P-HEOs and their local atomistic features. For a series of representative P-HEOs the short-range elemental distribution, chemical bond bending, oxygen-metal hybridization, and corresponding covalency visualized by real-space charge-density maps on sub-ångström length scales, are studied in this work. The composition fluctuations are directly observed in the elemental mapping, which may be considered as the first step in understanding of unique properties of P-HEOs. Of special interest for magnetic property-structure relationships are insights gained into composition-driven local oxygen octahedra distortions and their consequences for the hybridization of oxygen and transition-metal orbitals in P-HEOs. The observed structure–electronic phenomena, such as the variations in the bond covalency shown by the charge mapping, are related to the coexistence of complex magnetoelectronic states postulated in the previous reports on P-HEOs. The atomic-scale mapping of chemical distribution, lattice distortion and charge-density maps as well as the correlation of these structural characteristics with their physical (magnetic) properties may open another perspective and provide fundamental basis for the rational design of HEOs with fascinating properties.

## Methods

**Characterization.** Powder X-ray diffraction (XRD) measurements were conducted at room temperature using a STOE Stadi P diffractometer equipped with a Ga-jet X-ray source (Ga-Kβ radiation, 1.2079 Å). High-angle annular dark-field images (HAADF), annular bright-field (ABF), atomic-resolution X-ray energy-dispersive spectroscopy (EDS), and electron energy-loss spectroscopy (EELS) have been used to character the crystal structure, elemental distribution, TM–O-TM angle, O $K$-edge electron energy-loss near-edges structures of the four samples at the [110] zone axis, using a JEM300CF aberration-corrected scanning transmission electron microscope (STEM) operating at 300 keV with a beam current of 36 pA. STEM images were taken with the convergence angle of the incident electrons at 32 mrad and the collection angle at 90–165 mrad. EDS mappings were acquired using dual silicon-drift detectors (SDDs). In total, 50 scans (each with a 0.4 ms dwell time) in the same area were summed. A dispersion of 0.25 eV per channel was used and the

**Table 1 A summary of the average TM–O–TM bond angle, magnetic transition temperatures and O $K$ ELNES of the three P-HEOs.**

| P-HEOs | Average TM–O-TM bond angle (°) | Magnetic transition temperatures (K) | O $K$ ELNES (eV) |
|---|---|---|---|
| La(5TM$_{0.2}$)O$_3$ | 175.6 | 185 | 531.7 |
| (5RE$_{0.2}$)(5TM$_{0.2}$)O$_3$ | 153.4 | 135 | 531.9 |
| Gd(5TM$_{0.2}$)O$_3$ | 142.6 | 120 | 533.0 |

dwell time was 0.5 s per pixel for acquisition of EELS spectrum. The pre-edge background of the EELS spectrums was removed by a power-law function in DigitalMicrograph. The images were acquired from regions near the particle edges.

**STEM imaging method**. To enhance the signal-to-noise ratio of the STEM images of the samples, the images were filtered in Fourier space through introducing all the diffraction frequencies and smoothed using DigitalMicrograph (a commercial software). The horizontal drift in the charge-density map was adjusted by measuring the average displacement per line from a conventional fast-scanned STEM image and then shifting back. The vertical drift in the charge-density map was adjusted by rescaling the vertical axis to fit with the fast-scanned STEM image. DigitalMicrograph was also used to smooth the charge-density maps.

**Analysis of the elemental fluctuation from the EDS mapping**. To quantify the EDS elemental signal from the atomic columns, we use the A-site and B-site atomic positions and radius measured from HAADF-STEM images to define the integration region in the corresponding EDS maps. A-site and B-site atomic positions were measured by using two-dimensional (2D) Gaussian fitting on their atomic columns to locate the centers. The atomic radius was measured as the half of full-width-half-maximum of the 2D Gaussian fitting. With the region defined, we integrated the intensity inside the region and calculated the atomic fraction at each atomic column.

**Sample thickness measurement**. The thickness and orientation of the sample have big impacts on the PACBED patterns[50]. To estimate the sample thickness and align the sample to the zone axis, we employ PACBED patterns. For this scenario, we utilize a smaller convergence angle of 10.6 mrad. The sample tilt is more visible in this case than in regular imaging, making it easier to tilt the sample to the zone axis. Then, when the beam scans over the areas illustrated in Fig. 5a, we gathered the PACBED patterns by capturing the average CBED patterns in the diffraction plane. Finally, experimental PACBED patterns are compared with simulated ones with thicknesses 2–10 nm (generated using multi-slice simulation[51]) by using least-squares fitting to quantitatively determine the sample thickness (Supplementary Fig. 5). The best fit thickness is ~3.6 nm for $LaCoO_3$, $(5RE_{0.2})(5TM_{0.2})O_3$ and $Gd(5TM_{0.2})O_3$, and ~4.8 nm for $La(5TM_{0.2})O_3$ as highlighted in the black dashed rectangles. Since 4D-STEM data is only valid in thin regions (<6.6 nm)[25], our thickness measurements confirmed that the 4D-STEM results shown in Fig. 5a are reasonable.

**Charge calculation**. When an electron probe transmits through a material, it shifts in the diffraction plane in a way that is negatively proportional to the local electric field, due to the change in momentum caused by the electric field. The electric field is calculated based on the momentum change of electron beam. First, the dataset's center of mass (COM) was measured for each diffraction pattern. The deviation of the COM from the center of the diffraction pattern at each scanning position provides us a vector field. The pixel size should then be calibrated in momentum space (mrad/pixel). The momentum change of electrons is based on Eq. (3):

$$\frac{d\mathbf{p}}{dt} = -e\mathbf{E} \qquad (3)$$

We can rewrite the equation (as shown in Eq. 4) in terms of a few basic parameters based on the assumptions established in earlier studies[25,52,53].

$$\Delta \mathbf{p}_{xy} = -\frac{e\Delta z}{v_z}\mathbf{E}_{xy} \qquad (4)$$

Where $\Delta \mathbf{p}_{xy}$, $e$, $\Delta z$, $v_z$, and $\mathbf{E}_{xy}$ are the momentum chage, the charge of an electron, the sample thickness, the speed of the electron along the beam direction, and the electric field, respectively. Because the diffraction pattern is a momentum space image of the probe, $\Delta \mathbf{p}_{xy}$ can be calculated from the shift in the COM of the diffraction pattern (Eq. 5):

$$\Delta \mathbf{p}_{xy} = \Delta \mathbf{COM} p_z \qquad (5)$$

Where $\Delta \mathbf{COM}$ and $p_z$ are the shift in the COM from the geometric center of the diffraction pattern and the momentum of the electron beam along the beam direction, respectively. Therefore, we can calculate the electric field strength by Eq. (6):

$$\mathbf{E}_{xy} = -\frac{\Delta \mathbf{p}_{xy} v_z}{e\Delta z} = -\frac{\Delta \mathbf{COM}\, p_z v_z}{e\Delta z} \qquad (6)$$

Then, using Gauss's law, we can compute the electric field's divergence and calculate the charge density (Eq. 7):

$$\nabla \cdot \mathbf{E} = \frac{\rho}{\varepsilon_0} \qquad (7)$$

Where $\rho$ is the charge density and $\varepsilon_0$ is the vacuum permittivity. We eliminated the z dependency by integrating both sides along the z axis (beam direction) to suit the 2D charge-density map. The 4D-STEM data was collected with a Gatan OneView camera at a speed of 300 frames per second and a scanning step size of 0.2 Å.

## Data availability
The data that support the findings of this research are available within the article and the Supplementary Information file. All data are available on reasonable request from the corresponding authors.

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

## Acknowledgements

The research was primarily supported by the National Science Foundation (NSF) Materials Research Science and Engineering Center (MRSEC) program through the UC Irvine Center for Complex and Active Materials (DMR-2011967), and additional support received from the NSF under grant numbers DMR-2034738 and the Department of Energy, Office of Basic Energy Sciences, Division of Materials Sciences and Engineering under Grant DE-SC0014430. L.S. acknowledges support from the Research Plan for Graduate Mobility Program of Xi'an Jiaotong University. A.S. and H.H. acknowledge financial support from the Deutsche Forschungsgemeinschaft (DFG) under project HA 1344/43-2. The authors also acknowledge the use of facilities and instrumentation at the UC Irvine Materials Research Institute (IMRI) supported in part by the NSF's MRSEC program.

## Author contributions

X.P. and Ho.H. conceived and directed the project. L.S., Hu.H., and W.G. performed the STEM experiments and data analysis with the help of X.Y. and C.A. A.S. prepared the samples and did the XRD analysis with the help of R.K. L.S. and Hu.H. wrote the paper. All the authors discussed the results and commented on the manuscript.

## Competing interests

The authors declare no competing interests.
