## [Peer Review File · Nature Communications]

Title: Direct observation of elemental fluctuation and oxygen distortion-dependent charge distribution in high entropy oxidesREVIEWER COMMENTS

Reviewer #1 (Remarks to the Author):

Noteworthy results:

- Cations randomly fluctuate on the atomic scale for the mentioned HEO perovskites (ABO₃)
- BO₆ polyhedron distort locally based on the HEO local composition
- Re-HEO perovskites have larger local polyhedral distortions compared with TM.

These results show direct evidence, with atomic resolution, that the local structure is distorted in “high entropy” oxide perovskites. I would recommend the authors examine some of the literature on neutron scattering experiments and computational studies on perovskites. Those studies also show local structure distortions in simple perovskites (not HEOs). Comments/comparisons on that literature would bolster this work.

- <https://journals.aps.org/prb/abstract/10.1103/PhysRevB.91.024104>
- <https://doi.org/10.1103/PhysRevB.71.104430>
- <https://doi.org/10.1038/s41598-020-60475-8>

Here is a couple of other recent HEO reviews that were not mentioned:

- <https://doi.org/10.1063/5.0003149>
- <https://doi.org/10.1016/j.actamat.2020.10.043>
- <https://doi.org/10.1016/j.matdes.2021.109534>

The results clearly demonstrate the conclusions. The electron density maps clearly show BO₆ polyhedron distortions. The methodology is clear and allows reproducibility.

Minor comments:

Figure 1 is a little confusion. Are you trying to show that multiple component oxides introduce local ordering (nano-meter scale) and oxygen distortions? If so, please make it clearer. Instead of saying right and left, perhaps number?

Reviewer #2 (Remarks to the Author):

This manuscript reveals atomic scale structure information of a series of perovskite-high entropy oxides including short-range elemental distribution, chemical bond bending, oxygen-metal hybridization and corresponding covalency visualized by real space charge density maps at sub-ångström length scales by STEM for the first time. They also try to relate the structure information to magnetic properties indirectly. Overall, the direct examination of atomic structure of high entropy oxide is new and insightful but interpretation of the contrast is too straightforward. Besides, the structure and magnetic properties should be correlated more specifically in order to visualize more scientific impact of the derived

microstructure information. Therefore, to raise the quality of the work, I would suggest the authors to consider the following.

1. In supplementary Fig. 1, La(5TM0.2)O₃ does not seem to have the same diffraction peaks as the other two P-HEO materials. The phase identification should be confirmed. Besides, can the zig-zag bonding stacking be correlated with the XRD patterns?
2. In Fig. 2d, each element shows some degree of the brightness fluctuation as evidenced by the line profiles in Fig. 2e-2h. However, the peak of each element also fluctuates around each atomic column. Therefore, to locate the peak position followed by area integration over a radius might be more meaningful to represent the atomic fraction at each atomic column. Besides, the degree of fluctuation in atomic occupancy among all atomic columns might also be different among these 5 TM elements, which worth examining further to claim no obvious preference for specific neighbors of the TM cations.
3. Same comment to Gd(5TM0.2)O₃ in Supplementary Fig. 2c to g as above. Nevertheless, is there any reason causing the poorer contrast for imaging Gd(5TM0.2)O₃ than La(5TM0.2)O₃? Besides, why the atomic contrast of A sites is symmetric in La(5TM0.2)O₃ but asymmetric in Gd(5TM0.2)O₃?
4. In the case of (5RE0.2)(5TM0.2)O₃, not only the same fluctuation as mentioned before needs to be examined, but also the degree of deviation from the average composition of 0.2 for each element also differs, which also needs to be addressed and discussed.
5. Following above, in Fig. 3, EDX intensity can be found between atomic columns as interstitial sites for some elements. How to verify them to be real or not?
6. Is it possible to show atomic EDX oxygen map?
7. In Fig. 4, why the contrast of the O atomic columns in the ABF images varies a lot among these 4 samples? Can the lattice distortion induce different degree of diffraction contrast, which in turn affects the local crystal orientations and thus the shift the atomic column positions?
8. Would the alignment of the cation or anion ions in each atomic column affect the degree of blurring for the atomic contrast, representing fluctuation of atomic packing in three dimensions?
9. What is the physical meaning of MTF? Why MTF deviates from 1 only resulting in the anion ions (oxygen) shifting but not the cation ions to accommodate lattice strain?
10. Why the charge density maps of the negative contributions (indicated as red in Figure 5a) from both core and valence electrons show positive charge density and positive contributions (indicated in blue in Figure 5a) from atomic nuclei show negative charge density in the profiles?
11. It is better to obtain more specific microstructure information from the right shifts of the lowest energy O K-edge by matching with theoretical calculations.
12. It is better to provide more solid correlation between the strength of the hybridization of the 3d TM and 2p O orbitals and magnetic properties to testify the scientific importance of the atomic-scale structure information provided. The current correlation between the decrease in the magnetic transition temperature with the changes in the average TM-O-TM bond angles (175.6°, 153.4° and 142.6°, respectively) is vague and not specific.
13. English should be improved. For example, Figure 5b shows O Kedge and L_{2,3} peaks of all the other elements “expect” Y (L_{2,3} edges of Y are over 2000 eV).

Point-by-Point Response to the Reviewers' Comments

Title: Direct observation of elemental fluctuation and oxygen octahedral distortion-dependent charge distribution in high entropy oxides

Manuscript ID: NCOMMS-21-27269-T

Dear editor and reviewers,

Thank you for your consideration of our manuscript entitled "Direct observation of elemental fluctuation and oxygen octahedral distortion-dependent charge distribution in high entropy oxides" (NCOMMS-21-27269-T). The reviewers have provided us with constructive comments on our paper. Based on the comments, we have carefully revised the manuscript. A major improvement is that we included the new discussion of previous literature on structure distortions in simple perovskites, as suggested by reviewer #1. To address the concerns raised by reviewer #2, detailed discussions and revision have also been added to improve the correlation between structure and magnetic properties, and to clarify diffraction peaks, brightness fluctuations, variation in contrast, and charge density maps.

The overall quality of the work has been improved with your help. In the hope that you like it. Please see below for our detailed responses to each point. The modified text was indicated in red in the revised manuscript.

Yours sincerely,

Xiaoqing Pan

Reviewer #1 (Remarks to the Author):

Noteworthy results:

- Cations randomly fluctuate on the atomic scale for the mentioned HEO perovskites (ABO_3)

- BO_6 polyhedron distort locally based on the HEO local composition

- Re-HEO perovskites have larger local polyhedral distortions compared with TM.

These results show direct evidence, with atomic resolution, that the local structure is distorted in “high entropy” oxide perovskites. I would recommend the authors examine some of the literature on neutron scattering experiments and computational studies on perovskites. Those studies also show local structure distortions in simple perovskites (not HEOs). Comments/comparisons on that literature would bolster this work.

- <https://journals.aps.org/prb/abstract/10.1103/PhysRevB.91.024104>

- <https://doi.org/10.1103/PhysRevB.71.104430>

- <https://doi.org/10.1038/s41598-020-60475-8>

Response: Following your suggestion, we have carefully studied the literature related to the local structure distortions in simple perovskite oxides, from which the present work greatly benefits. Accordingly, the discussion about the local oxygen octahedral distortion is expanded in the revised manuscript.

Added discussion

Such local oxygen octahedral distortions have also been observed in simple perovskite oxides in previous studies through pair distribution function analysis of the neutron diffraction results^{35–37}. These local octahedral distortions show a close association with the physical properties of perovskite oxides. For instance, the interaction between local distortions induces dipole moments that could form a spontaneous polarization, thus resulting in ferroelectricity in $BaTiO_3$ ³⁶. Another example is $La_{0.60}Ca_{0.40}MnO_3$, where the local distorted oxygen octahedra results in the local anisotropic environment around Mn^{4+} , which leads to a small degree of local ferroelectricity without any obvious macroscopic polarity³⁷. Therefore, direct observation of local oxygen octahedral distortion and unraveling the relationship between chemical compositions and octahedral distortion in the P-HEOs would facilitate the designing and engineering of their local lattice structure and physical properties.

35. Fang, H., Wang, Y., Shang, S. & Liu Z.-K. Nature of ferroelectric-paraelectric phase transition and origin of negative thermal expansion in $PbTiO_3$. *Phys. Rev. B* **91**, 024104 (2015).

36. Culbertson, C. M., et al. Neutron total scattering studies of group ii titanates ($ATiO_3$, $A^{2+} = Mg, Ca, Sr, Ba$). *Sci. Rep.* **10**, 3729 (2020).

37. Rodriguez, E. E., Proffen, T., Llobet, A., Rhyne, J. J. & Mitchell J. F. Neutron diffraction study of average and local structure in $La_{0.5}Ca_{0.5}MnO_3$. *Phys. Rev. B* **71**, 104430 (2005).

Here is a couple of other recent HEO reviews that were not mentioned:

- <https://doi.org/10.1063/5.0003149>

- <https://doi.org/10.1016/j.actamat.2020.10.043>

- <https://doi.org/10.1016/j.matdes.2021.109534>

Response: We have added these references to reflect the whole background of the present work.

The results clearly demonstrate the conclusions. The electron density maps clearly show BO_6 polyhedron distortions. The methodology is clear and allows reproducibility.

Minor comments:

Figure 1 is a little confusion. Are you trying to show that multiple component oxides introduce local ordering (nano-meter scale) and oxygen distortions? If so, please make it clearer. Instead of saying right and left, perhaps number?

Response: We thank the reviewer for appreciating our work. We have revised Figure 1 and its caption as shown below to make it clearer, as shown below.

Figure 1. Illustration of the concept of the present study. **a.** The correlation between local elemental distribution, atomic structure, chemical bonding and magnetic properties. **b.** Schematic crystallographic structure of $LaCoO_3$. **c.** Rhombohedral $LaCoO_3$ parent structure (bottom left) with negligible oxygen octahedral distortion and orthorhombic model of high entropy oxide (top right) with uniformly distributed elements and oxygen octahedral distortion (as that derived from XRD pattern). **d.** Compositional fluctuation with nanometer scale ordering and composition-dependent local oxygen octahedral distortion of the high entropy perovskite oxide that we observed in the present study. In (c) and (d), most of oxygen atoms overlapped with A-site La atoms are omitted for visual clarity, and only those oxygen atoms belonging to the marked oxygen octahedra are present. Local chemical composition fluctuation of transition metal at each B-site atomic column gives rise to different degree of local oxygen octahedral distortion, as indicated by the varied TM-O-TM bonding angle marked between the B-site atomic columns and the oxygen ones.

Reviewer #2 (Remarks to the Author):

This manuscript reveals atomic scale structure information of a series of perovskite-high entropy oxides including short-range elemental distribution, chemical bond bending, oxygen-metal hybridization and corresponding covalency visualized by real space charge density maps at sub-ångström length scales by STEM for the first time. They also try to relate the structure information to magnetic properties indirectly. Overall, the direct examination of atomic structure of high entropy oxide is new and insightful but interpretation of the contrast is too straightforward. Besides, the structure and magnetic properties should be correlated more specifically in order to visualize more scientific impact of the derived microstructure information. Therefore, to raise the quality of the work, I would suggest the authors to consider the following.

1. In supplementary Fig. 1, La(5TM_{0.2})O₃ does not seem to have the same diffraction peaks as the other two P-HEO materials. The phase identification should be confirmed. Besides, can the zig-zag bonding stacking be correlated with the XRD patterns?

Response: We agree with the reviewer that the XRD pattern of La(5TM_{0.2})O₃ appears different from the other P-HEOs, (5RE_{0.2})(5TM_{0.2})O₃ and Gd(5TM_{0.2})O₃. However, the reason for this is not the different crystallographic structures but the variations in the lattice parameters, which are as follows:

La(5TM_{0.2})O₃ : $a = 5.4653(2) \text{ \AA}$, $b = 5.5104(5) \text{ \AA}$ and $c = 7.7431(6) \text{ \AA}$

(5RE_{0.2})(5TM_{0.2})O₃ : $a = 5.3610(5) \text{ \AA}$, $b = 5.5012(5) \text{ \AA}$ and $c = 7.6326(6) \text{ \AA}$

Gd(5TM_{0.2})O₃ : $a = 5.2894(2) \text{ \AA}$, $b = 5.5301(5) \text{ \AA}$ and $c = 7.5697(6) \text{ \AA}$

The significant decrease in the a and c parameters in (5RE_{0.2})(5TM_{0.2})O₃ and Gd(5TM_{0.2})O₃ directly relates to the increase in the metric distortion, which is often used as the parameter to describe the amount of the deviation of the given structure from its ideal cubic structure with the highest crystallographic symmetry ($Pm-3m$ for the RE-TM perovskite group). Hence, the metric distortion correspondingly indicates the increase in the orthorhombic nature of the sample. For an ideal cubic case, i.e., $Pm-3m$, the metric distortion should be 0. The structural details of P-HEOs and the equations used for calculation of the metric strain in a $Pbnm$ system are provided in one of our initial report on P-HEOs¹². The metric distortion value for the P-HEOs obtained from the Rietveld refinement of the XRD pattern are 0.198, 0.633 and 1.609 for La(5TM_{0.2})O₃, (5RE_{0.2})(5TM_{0.2})O₃ and Gd(5TM_{0.2})O₃, respectively. The variation in the metric distortion or the orthorhombic nature of the sample can also concluded from the XRD pattern as the super structure reflections arising from the orthorhombic distortion gets stronger in (5RE_{0.2})(5TM_{0.2})O₃ and Gd(5TM_{0.2})O₃ compared to La(5TM_{0.2})O₃. In fact, La(5TM_{0.2})O₃ is the closest to a cubic structure, hence the peak splitting induced by the difference between a , b and $c/\sqrt{2}$ parameters is significantly lower. Nevertheless, La(5TM_{0.2})O₃ cannot be fitted with a cubic setting as the super structure reflexes, although relatively weaker compared to other P-HEOs, are still present, as shown in Figure RR1 (a magnified version of **Supplementary Fig. 1.**) **Figure RR2**, drawn from the structural file obtained from the Rietveld refinement of the experimental XRD patterns of P-HEOs, is presented to better illustrate the differences

between the degree of the orthorhombic nature in the P-HEOs.

The zig-zag bond stacking, which is related to the TM-O-TM octahedral tilting, can also be observed from the structure obtained from the Rietveld refinements of the XRD pattern, as shown in Figure RR3. However, in addition to the resolution limitation in XRD (especially regarding the position of the oxygen ions) it should be noted that XRD provides an average picture of the TM-O-TM bond fluctuation. Hence, in case of P-HEOs containing several TM cations resulting in different TM-O-TM bonds, a local picture of the bond stacking is necessary.

Ref. 12 in the manuscript. Sarkar, A. et al. Rare earth and transition metal based entropy stabilized perovskite type oxides. *J. Eur. Ceram. Soc.* **38**, 2318–2327 (2018).

Supplementary Fig. 1. XRD patterns coupled with Rietveld refinements of LaCoO_3 and the three P-HEOs.

Figure RR1. Comparison between the XRD pattern of $\text{La}(5\text{TM}_{0.2})\text{O}_3$ with the cubic $Pm\text{-}3m$ structure. The presence of the super structure reflexes confirms the presence of an orthorhombic structure.

Figure RR2 (Supplementary Fig. 2). Structure model of the three P-HEOs obtained from the Rietveld refinement of the XRD pattern. **a to c** Structure model along [001] projection (view along the *c*-axis), of $\text{La}(5\text{TM}_{0.2})\text{O}_3$, $(5\text{RE}_{0.2})(5\text{TM}_{0.2})\text{O}_3$ and $\text{Gd}(5\text{TM}_{0.2})\text{O}_3$, respectively. **a' to c'** Structure model along [110] projection, of $\text{La}(5\text{TM}_{0.2})\text{O}_3$, $(5\text{RE}_{0.2})(5\text{TM}_{0.2})\text{O}_3$ and $\text{Gd}(5\text{TM}_{0.2})\text{O}_3$, respectively, showing the zig-zag TM-O-TM bond. The blue arrow shows the shift of A-site cations. The yellow line indicates the zig-zag TM-O-TM bond.

Added discussion:

In the XRD analysis part:

With the decrease of MTF, a significant decrease in the *a* and *c* lattice parameters in P-HEOs is observed (Supplementary Fig. 1). Likewise, with the decrease of the MTF, the super structure reflections get stronger, suggesting an increasing degree of orthorhombic distortion from $\text{La}(5\text{TM}_{0.2})\text{O}_3$ to $(5\text{RE}_{0.2})(5\text{TM}_{0.2})\text{O}_3$ to $\text{Gd}(5\text{TM}_{0.2})\text{O}_3$. Supplementary Fig. 2 shows the structure models of the three P-HEOs obtained from the Rietveld refinement of the XRD pattern, from which the differences between the degree of the orthorhombic nature in the P-HEOs can be observed.

In the ABF images analysis part:

...Although the oxygen octahedral distortion-induced zig-zag bond stacking can also be observed from the structure obtained from the Rietveld refinements of the XRD pattern (Supplementary Fig. 2), it should be noted that XRD provides only an average value of the TM-O-TM bond fluctuation. Hence, in case of P-HEOs containing several TM cations resulting in different TM-O-TM bonds, the local picture of the bond stacking observed through ABF imaging becomes important.

2. In Fig. 2d, each element shows some degree of the brightness fluctuation as evidenced by the line profiles in Fig. 2e-2h. However, the peak of each element also fluctuates around each atomic column. Therefore, to locate the peak position followed by area integration over a radius might be more meaningful to represent the atomic fraction at each atomic column. Besides, the degree of fluctuation in atomic occupancy among all atomic columns might also be different among these 5 TM elements, which worth examining further to claim no obvious preference for specific neighbors of the TM cations.

Response: According to the method that you suggested, we have updated the line profiles in the revised manuscript (as shown in revised Figure 2), and the detailed data processing method is added in the revised manuscript. The variation trends of the elemental fluctuation are very similar to the previous version. However, a clearer relative concentration of each element at each column is seen with such approach. Based on these data analysis, we compared the degree of fluctuations in atomic occupancy among the atomic columns, and further confirmed that there is no obvious preference for specific neighbors of the TM cations. We have updated the data analysis methods and added related discussion of the results in the revised manuscript.

Added methods:

Analysis of the elemental fluctuation from the EDS mapping: To quantify the EDS elemental signal from the atomic columns, we use the A site and B site atomic positions and radius measured from HAADF STEM images to define the integration region in the corresponding EDS maps. A site and B site atomic positions were measured by using two-dimensional (2D) Gaussian fitting on their atomic columns to locate the centers. The atomic radius was measured as the half of full-width-half-maximum of the 2D Gaussian fitting. With the region defined, we integrated the intensity inside the region and calculated the atomic fraction at each atomic column.

Added discussion:

The degree of fluctuation in atomic occupancy among all atomic columns is different among these 5 TM elements. For example, the atomic fractions of Cr, Mn, Fe, Co and Ni along the first column in Line 1 are 0.265, 0.205, 0.070, 0.318 and 0.142, respectively (Figure 2e), while those along the first column in Line 2 are 0.269, 0.185, 0.263, 0.173 and 0.110, respectively (Figure 2f). The atomic fraction of the TM cations between the neighboring columns changes irregularly (Figure 2e to h) with no obvious preference for specific neighbors.

Revised Figure 2. (a) HAADF-STEM, (b) ABF and (c) marked ABF images of $\text{La}(\text{5TM}_{0.2})\text{O}_3$ taken from $[110]$ zone axis. Scale bar, 0.5 nm. The green and silver dots marked in c denotes the B-site atoms and oxygen atom, respectively. (d) Atomic EDS mapping showing the disordered elemental distribution in the B site of the ABO_3 structure. (e to f) Line profiles of the atomic fraction calculated from the relative intensity, representing the distribution of individual B-site elements in a (001) plane projected along the $[110]$ zone axis.

3. Same comment to $Gd(5TM_{0.2})O_3$ in Supplementary Fig. 2c to g as above. Nevertheless, is there any reason causing the poorer contrast for imaging $Gd(5TM_{0.2})O_3$ than $La(5TM_{0.2})O_3$? Besides, why the atomic contrast of A sites is symmetric in $La(5TM_{0.2})O_3$ but asymmetric in $Gd(5TM_{0.2})O_3$?

Response: We have revised the line profiles of the atomic intensity in the EDS maps of $Gd(5TM_{0.2})O_3$ by using the method introduced above, as shown in Supplementary Fig. 3 in the revised Supplementary Information (Supplementary Fig. 2 in previous version, also shown below).

We believe the referee's question is about the asymmetric shape of the A-site column in STEM images. We agree with the reviewer that the atomic contrast of A sites is asymmetric, which is due to the overlap of A and O atoms at A site along $\langle 110 \rangle$ projection. Therefore, the atomic contrast at A site is combined from A atom and O atom (as shown in Supplementary Fig. S2a' to c'). Due to the larger O-octahedron rotation in $Gd(5TM_{0.2})O_3$ compared to $La(5TM_{0.2})O_3$, O atoms are not aligned in the same column with the A-site cations. Thus, the larger rotation in $Gd(5TM_{0.2})O_3$ causes high asymmetry in the column image.

Revised Supplementary Fig. 3. Atomic structure and compositional fluctuation of $Gd(5TM_{0.2})O_3$. **a** HAADF-STEM and **b** ABF of $Gd(5TM_{0.2})O_3$. Scale bar, 0.5 nm. **c** Atomic EDS mapping showing the disordered elemental distribution in the B site of $Gd(5TM_{0.2})O_3$. **d** to **g** Line profiles of the atomic intensity representing the distribution of individual B-site elements in a (001) plane projected along the [110] zone axis.

4. In the case of $(5RE_{0.2})(5TM_{0.2})O_3$, not only the same fluctuation as mentioned before needs to be examined, but also the degree of deviation from the average composition of 0.2 for each element also differs, which also needs to be addressed and discussed.

Response: According to the method that you suggested, we have updated the line profiles in Figure 3 in the revised manuscript. By using this method, the trend of the fluctuation of each element is similar to that was before, and average composition of Y and Ni in the profiles became closer to 0.2 than before, as shown below.

Revised Figure 3. Characterization of $(5RE_{0.2})(5TM_{0.2})O_3$. (a) HAADF and (b) ABF images of $(5RE_{0.2})(5TM_{0.2})O_3$ taken from $[110]$ zone axis. Scale bar is 0.5 nm. (c) Atomic EDS mapping showing the disordered elemental distribution in the B site of the ABO_3 structure. Line profiles of the atomic fraction calculated from the relative intensity fraction along each marked lines, representing the distribution of individual (d to f) A-site and (g to i) B-site elements in a (001) plane projected along the $[110]$ zone axis.

5. Following above, in Fig. 3, EDX intensity can be found between atomic columns as interstitial sites for some elements. How to verify them to be real or not?

Response: As the reviewer mentioned, EDX intensity can be found between atomic columns as interstitial sites for some elements. Such effect is caused by channeling effect rather than real signal. It is related to the dynamic diffraction of the incoming beam into the Bloch waves on certain atomic planes^{R1}. In our case, however, the channeling effect doesn't affect the results very much because our samples are relative thin (less than 5 nm, as displayed in Supplementary Fig. 4)^{R2}.

R1. Liao, Y. & Marks L. D. Reduction of electron channeling in EDS using precession.

Ultramicroscopy **126** 19–22 (2013).

- R2. Voyles, P. M., Grazul, J. L. & Muller, D. A. Imaging individual atoms inside crystals with ADF-STEM. *Ultramicroscopy* **96**, 251–273 (2003).

6. Is it possible to show atomic EDX oxygen map?

Response: We have provided here the EDS maps for oxygen. As shown in **Figure RR3**, Oxygen is seen between the B-site elements projected along the $\langle 110 \rangle$ direction, which corresponds well to the ABF images. However, because oxygen is a light element, we only see the oxygen lattice in the EDS map but not at atomic resolution.

Figure RR3. EDS mapping of one of the B-site element and the corresponding EDS maps of oxygen in the three high entropy perovskite oxides.

7. In Fig. 4, why the contrast of the O atomic columns in the ABF images varies a lot among these 4 samples? Can the lattice distortion induce different degree of diffraction contrast, which in turn affects the local crystal orientations and thus the shift the atomic column positions?

Response: Yes, we agree with the reviewer that the contrast of the O atomic columns in the ABF images displays slight change among these 4 samples, mainly due to changes in oxygen octahedral distortion. The broader distribution of oxygen octahedral distortion would lead to the increase of the projected area of the oxygen columns but a lower contrast^{R3}.

- R3. Gao, W., et al. Real-space charge-density imaging with sub-ångström resolution by four dimensional electron microscopy. *Nature* **575**, 480–484 (2019).

8. Would the alignment of the cation or anion ions in each atomic column affect the degree of blurring for the atomic contrast, representing fluctuation of atomic packing in three dimensions?

Response:

Conventionally, the degree of blurring during TEM imaging is usually affected by the sample thickness, image rotation, scanning noise, detector noise and aberrations. In the case of P-HEOs, besides the abovementioned factors, alignment of the cation

ions in each atomic column would be another factor that may influence the degree of blurring. As illustrated, elemental fluctuation in the P-HEOs results in the broad distribution of TM-O-TM angles, which would thus cause the deviation of ions from their ideal lattice structures, especially oxygen positions may not coincide with cations positions in projection, thus increasing the degree of blurring of the atomic columns. To clarify the effect of the alignment of the ions in each atomic column on the degree of blurring for the atomic contrast, we analyzed the relationship between the size of atomic columns and average value of its neighboring TM-O-TM angles in 20 groups. However, as shown in the figure below, the data distributes randomly without the trend that the atom size increases with the TM-O-TM bonding angle (oxygen octahedral rotations). Therefore, based on the present data, the alignment of cations or anions in each atomic column does not obviously affect the degree of blurring for the atomic contrast.

Figure RR4. relationship between the size of atomic columns and average value of its neighboring TM-O-TM angles.

9. What is the physical meaning of MTF? Why MTF deviates from 1 only resulting in the anion ions (oxygen) shifting but not the cation ions to accommodate lattice strain?

Response: The Goldschmidt's tolerance factor referred here as mean tolerance factor, MTF (t) is a theoretical parameter used for expressing the deviation of the ambient A-O and B-O equilibrium bond lengths from the ideal cubic structure ($Pm-3m$) of the perovskites. Therefore, the physical meaning of MTF is to appraise the degree of deviation of the lattice structure of perovskites from the ideal cubic structure, where $t \approx 1$ results in the $Pm-3m$ structure. For $t < 1$, i.e., for systems with smaller A-site cation or bigger B-site cations, orthorhombic or rhombohedral structures are preferred, whereas tetragonal or hexagonal structures are preferably formed for $t > 1$. The MTF in an oxide perovskite (ABO_3) is typically calculated from the ionic radii based on the following equation:

$$t = \frac{r_A + r_B}{\sqrt{2}(r_B + r_O)}$$

where r_A and r_B are the ionic radii of the cation at A-site and B-site, respectively, and r_O is the radius of the oxygen ion. In case of multiple cations at a specific site, like in P-HEOs, an average of the ionic radii is considered.

In case of the P-HEOs, $t < 1$, i.e., 0.96, 0.92 and 0.90 for $\text{La}(\text{5TM}_{0.2})\text{O}_3$, $(\text{5RE}_{0.2})(\text{5TM}_{0.2})\text{O}_3$ and $\text{Gd}(\text{5TM}_{0.2})\text{O}_3$, respectively. The mismatch in the bond lengths created by $t < 1$ is largely accommodated by cooperative rotations/tilt of the BO_6 octahedra. The Glazer tilt notation for orthorhombic perovskites with $Pbnm$ space group is $a^-a^+c^+$. The rotation of the BO_6 octahedra stems from the fact that A-site RE cations are too small for the cuboctahedral cage site, and so the BO_6 octahedra rotates/tilts to effectively reduce the cavity dimensions, thus allowing the structure to accommodate values of $t < 1$. The shifts in the oxygen position resulting from the octahedral rotations are accompanied by a corresponding shift of the A-site cations. In fact, the atomic site positions for the A-site cations is $(4a) x, y, 1/4$, which means that the x and y positions are variable and depends on the deviation of the MTF from ideality. Although the shift of the cations is less significant compared to the BO_6 octahedra tilting, it still needs to be considered during the Rietveld refinement of the XRD patterns, as shown in **Figure RR2a to c**.

Figure RR2 (Supplementary Fig. 2). Structure model of the three P-HEOs obtained from the Rietveld refinement of the XRD pattern. **a to c** Structure model along [001] projection (view along the c -axis), of $\text{La}(\text{5TM}_{0.2})\text{O}_3$, $(\text{5RE}_{0.2})(\text{5TM}_{0.2})\text{O}_3$ and $\text{Gd}(\text{5TM}_{0.2})\text{O}_3$, respectively. **a' to c'** Structure model along [110] projection, of $\text{La}(\text{5TM}_{0.2})\text{O}_3$, $(\text{5RE}_{0.2})(\text{5TM}_{0.2})\text{O}_3$ and $\text{Gd}(\text{5TM}_{0.2})\text{O}_3$, respectively, showing the zig-zag TM-O-TM bond. The blue arrow shows the shift of A-site cations. The yellow line indicates the zig-zag TM-O-TM bond.

10. Why the charge density maps of the negative contributions (indicated as red in

Figure 5a) from both core and valence electrons show positive charge density and positive contributions (indicated in blue in Figure 5a) from atomic nuclei show negative charge density in the profiles?

Response: Sorry for the misleading presentation of the profiles. The red and blue curves in the previous profiles only show by the charge distribution variation along the polylines (Top line and Bottom line) as indicated the charge density maps in Figure 4a. The positive and negative charge densities are indicated by the values in the y axis of charge density. We have used purple and grey curves in the revised Fig. 4a to refer to the top and bottom polylines respectively in the charge density mappings to avoid the misleading.

Revised Figure 5. Real-space charge mapping and EELS analysis of LaCoO₃ and the three P-HEOs. **a** Charge mapping of the three P-HEOs, indicating a decrease of the covalency and increase of the ionicity of TM-O bond. The colored lines in the line profiles of the charge distribution show the variation along the lines (Top line and Bottom line) as indicated in the charge maps. **b** EELS of LaCoO₃ and the three P-HEOs. **c** Detailed EELS of O K-edge electron energy loss near-edges structures of the four samples. The dash line in **c** is a guide to eye showing the right shift of the lowest energy O K peaks.

11. It is better to obtain more specific microstructure information from the right shifts of the lowest energy O K-edge by matching with theoretical calculations.

Response: We have learnt from the related references on the theoretical calculation and experimental studies on the O K-edge of transition metal oxides. These previous works help us understand the correlation between the microstructure information and the right shift of the lowest energy O K-edge. We have added related discussions to address your suggestion in the revised manuscript, as shown below.

Added discussion:

To provide further insight into the relationships between the chemical bond and oxygen octahedral distortion, electron energy loss spectroscopy (EELS) analysis was then conducted. Figure 5b shows O K-edge and $L_{2,3}$ peaks of all the other elements expect Y ($L_{2,3}$ edges of Y are over 2000 eV). In conventional RE-TM perovskites, the lowest energy O K-edge feature typically corresponds to the 3d TM and 2p O orbital overlap^{40,41,42}. Gaussian fitting is applied to find these peak positions, revealing that the significant difference between the 4 spectra is in a peak region from 528 to 534 eV. The lowest energy O K-edge peaks of LaCoO₃, La(5TM_{0.2})O₃, (5RE_{0.2})(5TM_{0.2})O₃, and Gd(5TM_{0.2})O₃ are located at 531, 531.7, 531.9, and 533 eV, respectively, showing a right shift. The shifts may stem from the changing of TM-O-TM bonding geometry or transition metal oxidation state, or both⁴⁰. In a previous work, researchers calculated the energy loss near-edge structures of LaFeO₃ and Bi_{0.9}La_{0.1}FeO₃ by density functional theory³⁹. Because the electronegativity of La (1.10) is only about half of that of Bi (2.02), less electrons are expected on the Fe and O sites in the case of Bi_{0.9}La_{0.1}FeO₃, resulting in the right shift of the lowest energy O K-edge peak. In another case, for a series of Mn oxides, the O K-edge increases from 528.02 to 532.00 eV when the Mn ions state changes from +4 to +2⁴¹. These examples have shown that the onset position of O K-edge increases with the decrease of TM oxidation state. For the present samples, the states of the RE cations in the A site and oxygen ions are +3 and -2, respectively, and the electronegativity of these RE elements (1.10, 1.14, 1.17, 1.20 and 1.22 for La, Nd, Sm, Gd and Y, respectively) is very close from one to another. Consequently, the average oxidation state of the TM cations in the B site does not change. Therefore, the shift of the O K-edge might only stem from the difference in the geometry of the TM-O-TM bond in the P-HEOs. Previous STEM, EELS and first-principles band structure calculation study on the electronic structure of layered double perovskite La₂CuSnO₆ has shown that oxygen octahedral distortions in the material would lead to an expansion of the Cu-O-Cu bond, resulting in the shift of the O K-edge position at the Cu site to a higher energy⁴². Another research on the electronic structure of perovskites CaCu₃Ir₄O₁₂, CaCu₃Ru₄O₁₂, and CaCu₃Ti₄O₁₂ also shows that the different B-site elements in the three oxides result in the difference of the TM-O (Cu-O and Ca-O) bond lengths, thus influencing the interaction between TM and oxygen⁴³. Therefore, the onset of O K-edge of CaCu₃Ti₄O₁₂ is located at the highest energy position among the three perovskite oxides. In the present case, with the increase of the TM-O-TM bond angles in the P-HEOs, the bond length of the TM-O-TM expands⁴². Consequently, the O K-edge position shift to higher energy position.

38. Ahn, C. C. Transmission Electron Energy Loss Spectrometry in Materials Science and the EELS Atlas. (Wiley-VCH Verlag GmbH & Co. KGaA Weinheim, 2004)
39. Sæterli, R., Selbach, S. M., Ravindran, P., Grande T. & Holmestad, R. Electronic structure of multiferroic BiFeO₃ and related compounds: Electron energy loss spectroscopy and density functional study. *Phys. Rev. B* **82**, 064102 (2010).
40. F. de Groot, F. M., et al. Oxygen 1s-x-ray absorption of tetravalent titanium oxides: A comparison with single-particle calculations. *Phys. Rev. B* **48**, 2074, 1993.
41. Tan, H., Verbeeck, J., Abakumov A. & Tendeloo, G. V. Oxidation state and chemical shift investigation in transition metal oxides by EELS. *Ultramicroscopy* **116**, 24–33 (2012).
42. Haruta, M., Kurata, H., Komatsu, H., Shimakawa, Y. & Isoda S. Site-resolved oxygen K-edge ELNES of the layered double perovskite La₂CuSnO₆. *Phys. Rev. B* **80**, 165123 (2009).
43. Xin, Y., Zhou, H.D., Cheng, J.G., Zhou, J.S. & Goodenough, J.B. Study of atomic structure and electronic structure of an AA'₃B₄O₁₂ double-perovskite CaCu₃Ir₄O₁₂ using STEM imaging and EELS techniques. *Ultramicroscopy* **127**, 94–99 (2013).

12. *It is better to provide more solid correlation between the strength of the hybridization of the 3d TM and 2p O orbitals and magnetic properties to testify the scientific importance of the atomic-scale structure information provided. The current correlation between the decrease in the magnetic transition temperature with the changes in the average TM-O-TM bond angles (175.6°, 153.4° and 142.6°, respectively) is vague and not specific.*

Response: Acknowledging the concerns raised by the reviewer we have tried to strengthen the physical connection between the local atomistic features and the magnetic properties in P-HEOs. We have already presented the details of magnetic features in P-HEOs in an earlier study²⁴. However, interpreting these magnetic properties requires understanding of the local atomistic aspects, which are inherently challenging to study in chemically complex P-HEOs. This study is an attempt to address this issue using the AC-STEM to get information on the local structure and chemical bonds. Yet, one needs to be cautious when ascribing a particular local atomistic aspect to a global magnetic response. The problem is that in P-HEOs any given measured magnetic characteristic is a joint effect of the several competing mechanisms. The three most important findings specifically related to the magnetic properties of P-HEOs are as follows:

- Scaling of the magnetic transition temperature with the means tolerance factor (MTF).
- Presence of vertical exchange bias (VEB), which is typical of heterogeneous magnetic systems, observed in single phase P-HEOs.
- Gradual magnetic transition happening over large temperature range.

Magnetic exchange interactions, which in case of the studied P-HEOs is predominantly superexchange antiferromagnetic (AFM), depend on the overlap of exponentially decaying orbital wave functions of the nearest-neighbor cations, in 3d-transition metals (TM) the 3d orbitals, that interact through the filled 2p-orbitals of the

neighboring oxygen. The exchange interaction between the participating ions in a simplified form can be described by the Heisenberg Hamiltonian, H , where J_{ab} is the exchange parameter coupling the two cations a and b, whose spins are S_a and S_b , respectively.

$$H = -2J_{ab}S_aS_b$$

The summation of all such binary metal ion interactions via oxygen (for instance in the frame of a mean field theory) determines the magnetic behavior of the material. The exchange parameter J_{ab} inherently depends on the degree of hybridization or the orbital overlap between the coupled ions, which in the considered cases is directly correlated with the bond angles. An increase in the degree of hybridization means increase in the covalent nature of the bond, which results in a stronger superexchange interaction. Consequently, stronger superexchange interactions lead to higher transition temperatures (Néel-temperature: T_N). Extensive studies on conventional orthorhombic RE-TM perovskites indicate that the interaction parameter J_{ab} is proportional to the cosine of the TM-O-TM bond angle (θ). Initially, it was proposed that J_{ab} is proportional to $\cos(\theta)$, while later studies reported that $\cos^2(\theta)$ directly provided a linear dependency of T_N ⁴⁸⁻⁵⁰. The most recent studies propose a modified correlation, i.e., J_{ab} is proportional to $\cos^4(\omega/2)/d^7$ where $\omega = 180^\circ - \theta$ and d = TM-O bond length⁴⁸. Given the value d is not constant in P-HEOs, as in single B-site perovskites, we have plotted T_N as a function of $\cos^2(\theta)$ in **Figure RR5**. As mentioned earlier, and also presented in the manuscript, the deviation of the TM-O-TM bond angle from 180° indicates lesser orbital overlap or decrease in the strength of hybridization. Hence, a decrease in the T_N is evident with lowering of the TM-O-TM bond angle. Consequently, the correlation between the T_N and MTF is also verified as the average TM-O-TM bond angle is directly related to MTF (as shown in Figure 4c in the main manuscript). For the information of the reviewer, we are presenting a comparison between the MTF, TM-O-TM bond angle and T_N in **Figures RR6**, where the additional data for some other P-HEOs (Nd(5TM_{0.2})O₃, Sm(5TM_{0.2})O₃ and Y(5TM_{0.2})O₃), whose bond angles are calculated from the Rietveld refinement of the XRD data, are included.

Figure RR5 (Supplementary Fig. 6). Correlation between the Néel-temperature, T_N , and the average TM-O-TM bond angle θ obtained from the HR-TEM analysis. A lowering of the T_N is observed with deviation of θ from 180° .

Figure RR6. Correlation between the (a) T_N vs. MTF and (b) T_N vs. TM-O-TM bond angle θ different P-HEOs. The bond angle for La(5TM_{0.2})O₃, (5RE_{0.2})(5TM_{0.2})O₃ and Gd(5TM_{0.2})O₃ are obtained from the HR-TEM analysis, while the ones for Nd(5TM_{0.2})O₃, Sm(5TM_{0.2})O₃ and Y(5TM_{0.2})O₃ are obtained from the Rietveld refinement of the XRD patterns.

Apart from the observed definitive variation in the transition temperature as a function of the TM-O-TM bond angle and resulting changes in the orbital hybridization, the presence of VEB and gradual change in transition temperature can also be elucidated based on the obtained local atomistic information. VEB fundamentally arises from the co-existence of ferromagnetism (FM) and antiferromagnetism (AFM) in a system, with Curie temperature of the FM phase being higher than the T_N of the AFM phase. Hence, the presence of the VEB necessarily indicates a magnetic phase-separation in crystallographic single-phase P-HEOs. While the superexchange AFM interactions are dominant in P-HEOs, it has been speculated that the presence of the FM phase/clusters is related to the preferential ferromagnetic exchange between some of TM cations. Although a definitive assignment of the FM interactions to certain local atomistic features is not straightforward, the current experimental observation of short-range elemental correlation on the TM B-sublattice in P-HEOs can be considered as one of the main factors. Likewise, the gradual onset of the magnetic transition over a large temperature regime can also be related to the chemically complex structure of the P-HEOs. The strength of the exchange interactions, which decides the magnetic transition temperature, are locally different in P-HEOs depending upon the local chemical compositions and distribution in the bond angles. Hence, instead of a sharp transition a gradual onset of magnetic ordering is observed in the P-HEOs. However, the exact reasons for the VEB and the gradual onset of magnetic transition are still hard to pin-point and further support from theoretical studies is necessary. Nevertheless, the current study at least provides us with some atomistic features which can primarily be associated with these unusual magnetic properties in P-HEOs.

24. Witte, R., et al. High-entropy oxides: An emerging prospect for magnetic rare-earth transition metal perovskites. *Phys. Rev. Mater.* **3**, 034406 (2019).

Revised discussion:

Correlation between the crystallographic structure and magnetic properties. The

crystallographic structure and the related TM-O-TM bond angle, constituent TM elements, TM-oxygen octahedral distortion and the extent of the $3d$ TM and $2p$ O overlap are of great significance for determining the magnetic properties of perovskite type oxides. In the previous work, it was postulated that the magnetic ordering of $\text{La}(\text{5TM}_{0.2})\text{O}_3$, which sets in gradually below 185 K, originates from the competing interactions between the predominant antiferromagnetic (AFM) matrix and nanometer length-scale weak ferromagnetic (FM) domains.²⁴ The presence and coupling between the two magnetic phases, enveloped in a single crystallographic structure, also resulted in an observable vertical exchange bias (VEB).²⁴ Speculations were made about the presence of short-range FM correlations stemming from the preferential ferromagnetic exchange interactions between some of transition metal cations.

Magnetic exchange interactions in ionic compounds, which in case of the studied P-HEOs is predominantly superexchange antiferromagnetic (AFM), depend on the overlap of exponentially decaying orbital wave functions of the nearest-neighbor cations, in $3d$ -transition metals (TM) the $3d$ orbitals, that interact through the filled $2p$ -orbitals of the neighboring oxygen. The exchange interaction between the participating ions in a simplified form can be described by the Heisenberg Hamiltonian, H (as indicated in equation 2), where J_{ab} is the exchange parameter coupling the two cations a and b , whose spins are S_a and S_b , respectively.

$$H = -2J_{ab}S_aS_b \quad (2)$$

The summation of all such binary metal ion interactions via oxygen (for instance in the frame of a mean field theory) determines the magnetic state of the material. The exchange parameter J_{ab} directly depends on the degree of hybridization or, in other words, on the orbital overlap between the neighboring ions. In the class of materials considered here the strength of hybridization is directly correlated with the bond angles. An increase in the degree of hybridization means increase in the covalent nature of the bond, which results in a stronger the superexchange interaction. Consequently, stronger superexchange interactions lead to higher transition temperatures (Néel-temperature: T_N). Extensive studies on conventional orthorhombic RE-TM perovskites indicates that the interaction parameter J_{ab} is proportional to the cosine of the TM-O-TM bond angle (θ). Initially, it was proposed that J_{ab} is proportional to $\cos(\theta)$, while later studies reported that $\cos^2(\theta)$ directly provided a linear dependency of T_N ⁴⁸⁻⁵⁰. The most recent studies propose a modified correlation, i.e., J_{ab} is proportional to $\cos^4(\omega/2)/d^7$ where $\omega = 180^\circ - \theta$ and $d = \text{TM-O bond length}$ ⁴⁸. Given the value d is not constant in P-HEOs, as in single B-site perovskites, we have plotted T_N as a function of $\cos^2(\theta)$ in Supplementary Fig. 6. As illustrated by the charge density maps and EELS analysis, the deviation of the TM-O-TM bond angle from 180° indicates lesser orbital overlap or decrease in the strength of hybridization. Hence, as displayed in Table 1, a decrease in the T_N from $\text{La}(\text{5TM}_{0.2})\text{O}_3$ to $(\text{5RE}_{0.2})(\text{5TM}_{0.2})\text{O}_3$ to $\text{Gd}(\text{5TM}_{0.2})\text{O}_3$ (from 185 K to 135 K to 120 K, respectively) is in complete agreement with the changes in the average TM-O-TM bond angles (175.6° , 153.4° and 142.6° , respectively), alterations indicated by the zig-zag lines in the charge

density maps and the red shift in the lowest energy feature of the O K ELNES feature (531.7, 531.9, and 533.0 eV, respectively) directly observed in the present work.

Essentially, the magnetic characteristics of P-HEOs can be broadly classified in two categories. One is the generic feature, such as the onset temperature of magnetic phase transition, which is strongly dependent on the overall crystallographic structure, the average bond angle, the zig-zag alteration in the charge density features and the resulting extent of the 3d TM and 2p O hybridization. On the other side, the distinctive magnetic features, such as the gradual magnetic phase transition distributed over a range of temperature and magnetic phase separation resulting in VEB are largely dictated by the local structural features, for instance, the broad distribution of TM-O-TM bond angles and the elemental fluctuations in P-HEOs. **The presence of the VEB necessarily indicates a magnetic phase-separation in crystallographic single-phase P-HEOs. While the superexchange AFM interactions are dominant in P-HEOs, it is plausible that the presence of the FM phase/clusters is related to the preferential ferromagnetic exchange between some of the TM cations. Although a definitive assignment the FM interactions to certain local atomistic feature is not straightforward, the current experimental observation of elemental fluctuation on the TM B-sublattice in P-HEOs can be considered as one of the main factors (Figures 2 and 3). Likewise, the gradual onset of the magnetic transition over a large temperature regime can also be related the chemically complex structure of the P-HEOs. The strength of the exchange interactions, which decides the magnetic transition temperature, are locally different in P-HEOs depending upon the local chemical compositions (Figures 2 and 3) and broad distribution in the bond angles (Figures 4b and c). Hence, instead of a sharp transition, a gradual onset of magnetic ordering is observed in the P-HEOs. Altogether, by utilizing the strength of the detailed STEM atomic-scale characterization techniques, it can be shown that the unique magnetic properties of P-HEOs are a direct consequence of their, especially local, structure and electronic distinctiveness.**

48. Zhou, J.-S. & Goodenough, J. B. Intrinsic structural distortion in orthorhombic perovskite oxides. *Phys. Rev. B* **77**, 132104 (2008).

49. Zhou, J.-S. & Goodenough, J. B. Unusual evolution of the magnetic interactions versus structural distortions in RMnO₃ perovskites. *Phys. Rev. Lett.* **96**, 247202 (2006).

50. Treves, D., Eibschütz, M. & Coppens, P. Dependence of superexchange interaction on Fe³⁺-O²⁻-Fe³⁺ linkage angle. *Phys. Lett.* **18**, 216–217 (1965).

13. *English should be improved. For example, Figure 5b shows O Kedge and L2,3 peaks of all the other elements “expect” Y (L2,3 edges of Y are over 2000 eV).*

Response: We have carefully checked the manuscript and corrected the typos in the manuscript as indicated in Red in the revised manuscript.

REVIEWERS' COMMENTS

Reviewer #1 (Remarks to the Author):

The authors have addressed all my major comments.

Reviewer #2 (Remarks to the Author):

The authors have addressed most of the concerns and raised the presentation quality significantly. No further comments.